# Learning mechanical systems from real-world data using discrete forced Lagrangian dynamics

## Abstract

We introduce a data-driven method for learning the equations of motion of mechanical systems directly from position measurements, without requiring access to velocity data. This is particularly relevant in system identification tasks where only positional information is available, such as motion capture, pixel data or low-resolution tracking. Our approach takes advantage of the discrete Lagrange-d'Alembert principle and the forced discrete Euler-Lagrange equations to construct a physically grounded model of the system's dynamics. We decompose the dynamics into conservative and non-conservative components, which are learned separately using feed-forward neural networks. In the absence of external forces, our method reduces to a variational discretization of the action principle naturally preserving the symplectic structure of the underlying Hamiltonian system. We validate our approach on a variety of synthetic and real-world datasets, demonstrating its effectiveness compared to baseline methods. In particular, we apply our model to (1) measured human motion data and (2) latent embeddings obtained via an autoencoder trained on image sequences. We demonstrate that we can faithfully reconstruct and separate both the conservative and forced dynamics, yielding interpretable and physically consistent predictions.

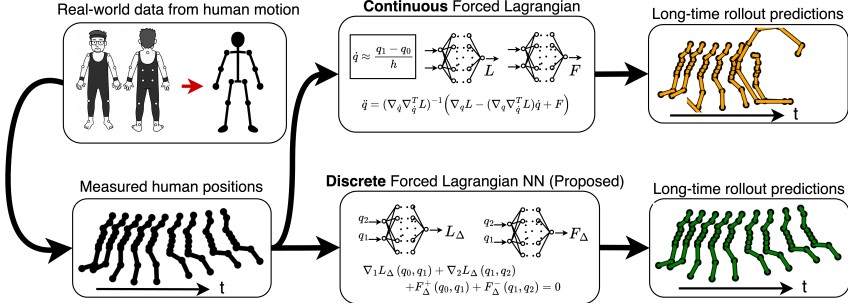

Figure 1: In this paper, we propose a structure-preserving approach for learning non-conservative system that directly learns from position data only (i.e., does not require velocities or momenta).

## 1 Introduction

Incorporating physics principles into machine learning has become a popular strategy for system identification and accurately predicting dynamics. Seminal examples are the Hamiltonian and Lagrangian neural networks, which are designed to preserve structure when applied to canonical

Submitted to 39th Conference on Neural Information Processing Systems (NeurIPS 2025). Do not distribute.

systems [1, 2]. These architectures have naturally been extended to include dissipation, enabling accurate modeling of realistic mechanical systems with physically motivated inductive biases [3, 4]. Such models offer advantages over traditional "black-box" modeling strategies that face challenges such as noise sensitivity, data sparsity, poor generalization, and limited interpretability [5, 6, 7, 8, 9]. Incorporating physics into the training processes is becoming an increasingly popular way to increase model generalization while giving physically interpretable predictions.

However, almost all known methods for learning dynamics rely on being able to observe momentum or velocity data, in addition to position data. In practice, velocity and momentum estimates are usually approximated from sequential position measurements using finite differences, which leads to inaccuracies due to noise and truncation errors [10]. Two examples considered in this paper are: (1) learning from pixel data or (2) motion tracking data, where instantaneous velocities are unavailable.

Building on ideas of geometric mechanics and variational integrators, [11], we propose a structure-preserving approach that directly learns from *position data only* (i.e., does not require velocities or momenta) and naturally incorporates dissipation and other non-conservative forces such as control terms. The method, which we refer to as a Discrete Forced Lagrangian Neural Network (DFLNN), is based on the discrete Lagrange-d'Alembert principle, which naturally reduces to a symplectic (variational) integrator on the configuration manifold in absence of non-conservative forces.

In a number of synthetic problems, the DFLNN is demonstrated to yield significant advantages over a non-structure-preserving neural ODE baseline model and, notably, a structure-preserving continuous analogue of the proposed method based on the (continuous) forced Euler-Lagrange equations where velocities are approximated using finite differences.

Additionally, we emphasize the flexibility of combining the proposed model with an autoencoder trained on images of damped pendulum dynamics. We show that we can separate the dissipative forces in the latent space to accurately recover and simulate pixel sequences of a *conservative* pendulum, despite the model only learning from the dissipative pixel images. Finally, we apply the model to real-world 3D human motion skeleton tracking data. The model successfully captures the underlying physics and shows excellent prediction performance over long times. Human motion data is extensively employed in simulations, animations, and biomechanical research [12]. Examining these movements has promising applications across various domains, including healthcare [13, 14] and sports [15]. Ensuring that models generalize effectively to unseen scenarios is crucial, particularly in medical applications.

Table 1: Comparison to other models. Models marked with a ≈ denotes approximate preservation of structure
.

|  | HNN [1] | LNN [2] | LSI [16] | D-HNN [17, 18] | GLNN [3] | NODE [19] | DFLNN (Ours) |
|---|---|---|---|---|---|---|---|
| (a) Learns from position only |  |  | ✓ |  |  | ✓ | ✓ |
| (b) Structure-preserving | ≈ | ≈ | ✓ | ≈ | ≈ |  | ✓ |
| (c) Incorporates dissipation |  |  |  | ✓ | ✓ |  | ✓ |

## 1.1 Related work

Understanding and modeling mechanical systems from data is vital in fields such as robotics, biomechanics, and structural health monitoring. While conventional system identification often assumes access to fully known physics, recent efforts focus on physics-informed learning, leveraging partial physical knowledge to improve generalization and interpretability [20, 21, 22, 23, 24, 16, 25, 26, 27, 28, 29, 30, 31]. Early approaches like Neural ODEs [19] and sparse identification of nonlinear dynamics [32] rely on black-box models or predefined basis functions. These have been paired with auto-encoders to identify low-dimensional latent ODE models from pixel data [33], but often neglect structural constraints such as conservation laws.

To address this, recent work embeds physical priors directly into the learning process. Hamiltonian and Lagrangian-based learning methods learn energy-based formulations that define the equations of motion, yielding models with physically consistent inductive biases [1, 34, 35, 36, 37, 38, 39, 2, 8, 40, 41, 42, 43, 44, 16]. When paired with symplectic or variational integrators, these networks can

preserve geometric properties such as the symplectic form or modified energy over long trajectories [26, 30, 27, 25, 29]. More recently, extensions to non-conservative systems have emerged, combining structure-preserving principles with dissipative or driven dynamics [37, 45, 3, 46, 47]. These approaches mark a shift toward hybrid models that respect both physical structure and the realities of real-world data.

## 1.2 Our contribution.

Our main contributions are now summarized, with a comparison to other models in Table 1.

a) We propose a discrete forced Lagrangian neural network that allows us to learn forced, dissipative dynamics from position data only, it does not require instantaneous velocity or momentum observations.

b) In the absence of external forces, our method reduces to a variational discretization of the action principle naturally preserving the symplectic structure of the underlying Hamiltonian system.

c) As it is based on the Lagrange-d'Alembert principle the DFLNN yields physically interpretable predictions, allowing us to separate conservative from non-conservative dynamics.

d) We show the method generalizes well on both synthetic and real-world data, as demonstrated in the computed predictions rollouts reproducing human motion.

e) We show that by implementing the DFLNN on learned autoencoder embeddings from pixel data, we can separate dissipative dynamics from the conservative dynamics in *latent* space.

## 2 Background

**Continuous d'Alembert Principle and Forced Euler-Lagrange Equations** In the presence of non-conservative forces, it is useful to consider the generalized formulation of Hamilton's principle, commonly referred to as the Lagrange-d'Alemberts principle. Let $Q$ denote a configuration manifold with $TQ$ its tangent bundle. The Lagrange-d'Alemberts principle seeks a curve $q : [t_0, t_N] \rightarrow \mathbb{R}^d$ that satisfies:

$$\delta \left( \int_{t_0}^{t_N} L\left(q(t), \dot{q}(t)\right) dt \right) + \int_{t_0}^{t_N} F\left(q(t), \dot{q}(t), u(t)\right) \cdot \delta q(t)\, dt = 0, \tag{1}$$

where $\delta$ denotes variations that vanish at the endpoints $q(t_0) = q_0$ and $q(t_N) = q_N$, $L : TQ \rightarrow \mathbb{R}$ is the Lagrangian function and $F : TQ \rightarrow TQ^*$ is the Lagrangian force, which is a fiber preserving map $F : (q, \dot{q}) \mapsto (q, F(q, \dot{q}))$, [11, 421]. This principle is equivalent to the *forced Euler-Lagrange equations*,

$$0 = \mathcal{E}(L, F)(q, t) := \frac{\partial L}{\partial q}\left(q(t), \dot{q}(t)\right) - \frac{d}{dt}\left(\frac{\partial L}{\partial \dot{q}}\left(q(t), \dot{q}(t)\right)\right) + F\left(q(t), \dot{q}(t)\right). \tag{2}$$

In absence of non-conservative forces, (1) expresses the extremization of the action functional, realizing Hamilton's principle, and (2) are the Euler-Lagrange equations. In previous work [3] Xiao et al. formulated their learning problem starting from equations (2), and proposed the Generalised Lagrangian Neural Networks (GLNNs), see also [2]. In their framework, $L$ and $F$ are functions approximated by neural networks using position and velocity data $(q, \dot{q})$ observed at discrete times; see [3], and Appendix C for details on this method and our implementation.

Since we are addressing scenarios where only position data are observed, following [11, 48], we propose instead to discretize the Lagrange d'Alembert principle, and derive discrete equations directly from the discrete principle.

**Discrete d'Alembert Principle and Forced Euler-Lagrange Equations** A discretization of (1) leads to the Discrete Lagrange-d'Alembert principle [11, 49]. Consider a partition of $[t_0, t_N]$ in $N$ subintervals of equal size, $h$. Consider a choice of numerical approximations of $F$ and $L$ on each

subinterval $[t_n, t_{n+1}]$, denoted by

$$L_\Delta(q_n, q_{n+1}) \approx \int_{t_n}^{t_n+h} L\big(q(t), \dot{q}(t)\big)\, dt,$$

$$F_\Delta^+(q_n, q_{n+1}, h) \approx \int_{t_n}^{t_n+h} F\big(q(t), \dot{q}(t)\big) \cdot \frac{\partial q(t)}{\partial q_{n+1}}\, dt, \qquad (3)$$

$$F_\Delta^-(q_n, q_{n+1}, h) \approx \int_{t_n}^{t_n+h} F\big(q(t), \dot{q}(t)\big) \cdot \frac{\partial q(t)}{\partial q_n}\, dt.$$

Here, $L_\Delta : Q \times Q \to \mathbb{R}$ and $F_\Delta^\pm : Q \times Q \to T^*Q$ represent the discrete Lagrangian and the discrete force, respectively. The discrete Lagrange-d'Alembert principle (dLDA) seeks a discrete trajectory, $\{q_n\}_{n=1}^N$, that satisfies the following condition:

$$\delta \sum_{n=0}^{N-1} L_\Delta(q_n, q_{n+1}) + \sum_{n=0}^{N-1} \big[ F_\Delta^-(q_n, q_{n+1}) \cdot \delta q_n + F_\Delta^+(q_n, q_{n+1}) \cdot \delta q_{n+1} \big] = 0, \qquad (4)$$

for all variations $\{\delta q_n\}_{n=0}^N$ vanishing at the endpoints $\delta q_0 = \delta q_N = 0$. Equivalently, (4) can be expressed in terms of the discrete forced Euler-Lagrange equations:

$$0 = \mathcal{E}_\Delta(L_\Delta, F_\Delta)(q_{n-1}, q_n, q_{n+1}) := \nabla_2 L_\Delta(q_{n-1}, q_n) + \nabla_1 L_\Delta(q_n, q_{n+1})$$
$$+ F_\Delta^+(q_{n-1}, q_n) + F_\Delta^-(q_n, q_{n+1}), \quad n = 2, \dots, N-1, \qquad (5)$$

where $\nabla_1$ and $\nabla_2$ denote differentiation with respect to the first and second variable.

## 3 Discrete Forced Lagrangian Neural Networks

The proposed model learns the dynamics of an observed system through neural network approximations of the Lagrangian and external forces,

$$L_\theta \approx L \quad L_\theta : TQ \to \mathbb{R}, \qquad F_\theta \approx F, \quad F_\theta : TQ \to TQ^*,$$

where $L_\theta$ and $F_\theta$ are parameterized with learnable parameters $\theta$.

The loss function minimized during training combines a physics term and a regularization term:

$$\mathcal{L} = \frac{\omega_{\text{physics}}}{N_\mathcal{T}(N+1)} \sum_\mathcal{T} \sum_{n=1}^{N-1} \mathcal{L}_{\text{physics}}(L_\theta, F_\theta)(q_{n-1}, q_n, q_{n+1}) + \frac{\omega_{\text{reg}}}{R} \sum_{r=1}^R \mathcal{L}_{\text{reg}}(L_\theta)(q_r, q_{r+1}). \qquad (6)$$

Here, $\mathcal{L}_{\text{physics}}$ is deduced from the discrete forced Euler-Lagrange equations (5), while $\mathcal{L}_{\text{reg}}$ promotes the regularity of $L_\theta$ [11, 379], each weighted with the hyperparameters $\omega_{\text{physics}}$ and $\omega_{\text{reg}}$. The dataset $\mathcal{T}$ contains $N_\mathcal{T}$ trajectories of $N$ steps each, and $R$ denotes the number of point pairs $(q_n, q_{n+1})$ used for regularization. The regularization may be applied over all such pairs in the training dataset. However, in the experiments, we demonstrate that it is sufficient to regularize only over a moderate number of pairs of data points.

As (6) only evaluates $\mathcal{L}_{\text{physics}}$ on local triplets $(q_{n-1}, q_n, q_{n+1})$, training is performed independently on such segments, without leveraging longer-range temporal dependencies present in the full trajectories. For generalizations of this approach using longer trajectory segments see Appendix A.

### 3.1 Learning Physics from the Observed Data ($\mathcal{L}_{\text{physics}}$)

Usually, the numerical discretization of a forced Lagrangian system would suggest for the exact $L$ and $F$ to be known, while the discrete trajectories $\{q_n\}_{n=0}^N$ are the unknowns of the problem. In the inverse setting considered here, the roles are reversed: the trajectories $\{q_n\}_{n=0}^N$ are given, and the goal is to recover $(L_\theta, F_\theta)$ as approximations to the underlying Lagrangian and forces.

A discretization scheme can be applied to obtain the discrete Lagrangian and discrete force $(L_\Delta, F_\Delta)$ as a function of $(L_\theta, F_\theta)$, (3). A concrete example using a mid-point approximation is

$$L_\Delta(q_n, q_{n+1}) := h\, L_\theta\left( \frac{q_n + q_{n+1}}{2}, \frac{q_{n+1} - q_n}{h} \right) = h\, L_\theta\left( \bar{q}_{n+\frac{1}{2}}, \dot{\bar{q}}_{n+\frac{1}{2}} \right) \qquad (7)$$

$$F_\Delta^\pm(q_n, q_{n+1}, h) := \frac{h}{2}\, F_\theta\left( \frac{q_n + q_{n+1}}{2}, \frac{q_{n+1} - q_n}{h} \right) = \frac{h}{2}\, F_\theta\left( \bar{q}_{n+\frac{1}{2}}, \dot{\bar{q}}_{n+\frac{1}{2}} \right) \qquad (8)$$

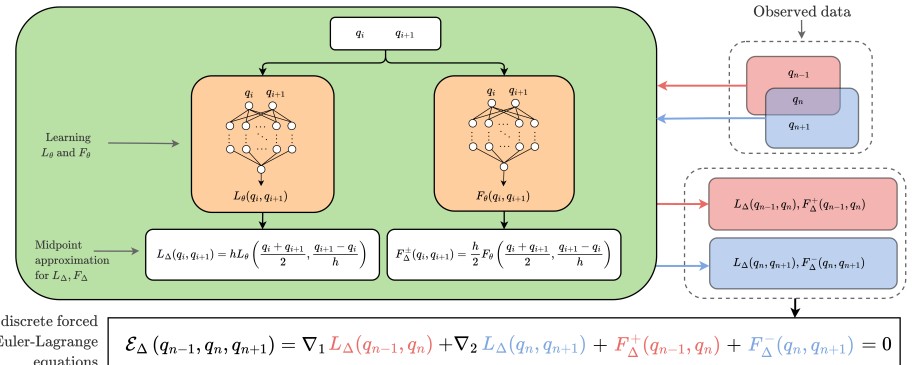

Figure 2: A schematic illustration of the data flow through the proposed model. We use segments of observed $(q_{n-1}, q_n, q_{n+1})$ to evaluate the discrete forced Euler-Lagrange equations.

where we have introduced the notation $\bar{q}_{n+\frac{1}{2}} := \frac{1}{2}(q_n + q_{n+1})$, $\bar{\dot{q}}_{n+\frac{1}{2}} := \frac{1}{h}(q_{n+1} - q_n)$ for brevity. We will use this discretization in our numerical experiments.

Assuming that the observed position data satisfy the dLDA principle (4) and, equivalently, the dLDA equations (5), a schematic illustration considering the flow of an observed trajectory of data points through the neural networks $L_\theta$ and $F_\theta$—used to construct the dLDA equations in (5)—is provided in Figure 2. Summing over all observed trajectories, we define the following physics term for the loss function:

$$
\begin{aligned}
\mathcal{L}_{\text{physics}}(L_\theta, F_\theta)(q_{n-1}, q_n, q_{n+1}) = \frac{h}{2} \bigg\| & \nabla_1 L_\theta\left(\bar{q}_{n-\frac{1}{2}}, \bar{\dot{q}}_{n-\frac{1}{2}}\right) + \frac{2}{h}\nabla_2 L_\theta\left(\bar{q}_{n-\frac{1}{2}}, \bar{\dot{q}}_{n-\frac{1}{2}}\right) \\
& + \nabla_1 L_\theta\left(\bar{q}_{n+\frac{1}{2}}, \bar{\dot{q}}_{n+\frac{1}{2}}\right) - \frac{2}{h}\nabla_2 L_\theta\left(\bar{q}_{n+\frac{1}{2}}, \bar{\dot{q}}_{n+\frac{1}{2}}\right) \quad (9) \\
& + F_\theta\left(\bar{q}_{n-\frac{1}{2}}, \bar{\dot{q}}_{n-\frac{1}{2}}\right) + F_\theta\left(\bar{q}_{n+\frac{1}{2}}, \bar{\dot{q}}_{n+\frac{1}{2}}\right) \bigg\|_2.
\end{aligned}
$$

### 3.2 The Learned Lagrangian ($L_\theta$)

The Lagrangian function $L_\theta$ can be learned either as a generic function or with embedded physical structure. In the most general case $L_\theta$ can be a feed-forward neural network. Alternatively, structural priors can be imposed. Assuming the Lagrangian takes a mechanical form, we can let

$$
L_\theta(\bar{q}_k, \bar{\dot{q}}_k) = \bar{\dot{q}}_k^T M_\theta(\bar{q}_k)\bar{\dot{q}}_k - U_\theta(\cdot), \qquad k = n \pm \frac{1}{2} \tag{10}
$$

with $M_\theta(\bar{q}_k) = \epsilon\mathbb{I} + \Lambda_\theta^T(\bar{q}_k)\Lambda_\theta(\bar{q}_k)$ ensuring $M_\theta$ is symmetric and positive definite. Here, $\Lambda_\theta : \mathbb{R}^d \to \mathbb{R}^{d \times d}$ is a lower triangular matrix parameterized by feed-forward neural network, and $\epsilon > 0$ a small constant ensuring the strict positiveness on $M_\theta$. $U_\theta(\cdot)$ is the potential energy function, also modeled as a feed-forward neural network, and may depend on $\bar{q}_k$ and/or $\bar{\dot{q}}_k$.

**Maximizing the regularity of the Lagrangian** Learning $L_\theta$ by minimizing (9) can lead to trivial solutions, preventing the discovery of meaningful physical relationships (e.g., the network might learn $L_\theta$ as a constant, causing the derivative terms in (9) to vanish without properly fitting the data). This is a well-known challenge in data-driven Lagrangian-based models, and there are different strategies proposed to address this problem [44, 16, 50]. A Lagrangian is regular, or non-degenerate, if and only if its Hessian with respect to the second argument is invertible, [11, 379]. We will require the invertibility to hold point-wise: specifically on a selected number of pairs of points $(\bar{q}_{r+\frac{1}{2}}, \bar{\dot{q}}_{r+\frac{1}{2}})$, we require that

$$
S(\bar{q}_{r+\frac{1}{2}}, \bar{\dot{q}}_{r+\frac{1}{2}}) := \left( \frac{\partial^2 L(\bar{q}_{r+\frac{1}{2}}, \bar{\dot{q}}_{r+\frac{1}{2}})}{\partial \bar{\dot{q}}_{r+\frac{1}{2}}^2} \right), \tag{11}
$$

156 is invertible. Here $\bar{q}_{r+\frac{1}{2}}$ and $\bar{\dot{q}}_{r+\frac{1}{2}}$ are as defined in Section 3.1 and depend on the data $(q_r, q_{r+1})$.
157 We then include in the loss function a regularization term $\mathcal{L}_{\text{reg}}$ as a logarithmic barrier that maximizes
158 the regularity of $S$ by penalizing the absolute value of its determinant [44],

$$\mathcal{L}_{\text{reg}}(q_r, q_{r+1}) = |\log\left(|\det(S(\bar{q}_{r+\frac{1}{2}}, \bar{\dot{q}}_{r+\frac{1}{2}})|\right)|. \tag{12}$$

### 3.3 The Learned External Force ($F_\theta$)

160 To learn the dynamics of systems with completely unknown forces, we parameterise $F_\theta = F_\theta^{\text{Free}}$ by a
161 neural network. To prevent $F_\theta^{\text{Free}}$ from learning conservative dynamics described by the Lagrangian
162 term, we apply dropout regularization [51]. This encourages slower learning of the force term and
163 encourages the Lagrangian term to fit the data when possible.

164 When prior knowledge suggests a dissipative structure, we instead let $F_\theta$ be a Rayleigh dissipation
165 function[52],

$$F_\theta(\bar{q}_k, \bar{\dot{q}}_k) = -K_\theta(\bar{q}_k)\,\bar{\dot{q}}_k, \qquad k = n \pm \frac{1}{2} \tag{13}$$

166 with $K_\theta(\bar{q}_k) = A_\theta^T(\bar{q}_k)A_\theta(\bar{q}_k)$ parameterized by a Cholesky factorization with the lower triangular
167 matrix $A_\theta$ to ensure positive-semi-definiteness. For linear dissipation, $F_\theta$ is independent of $\bar{q}_k$.

## 4 Experiments

169 To demonstrate that the proposed model can learn and separate the conservative and non-conservative
170 components of the observed system, we perform the following experiments: we train on data obtained
171 from non-conservative systems, but after training, we turn off the learned external force term $F_\theta$, and
172 evaluate the rollout predictions for the corresponding conservative system. We will present the results
173 obtained from the proposed model using a specific configuration of $L_\theta$ and $F_\theta$ architectures described
174 in the previous section. Alternative configurations, based on the assumed priors, are presented in the
175 appendix.

176 The model is evaluated in four learning tasks. Tasks 1 through 3 involve a damped double pendulum,
177 a dissipative charged particle moving within a magnetic field, and pixel frames from a damped
178 pendulum. Training data for these tasks are generated by solving the corresponding analytical ODEs
179 with Gaussian noise applied to each sample. In task 4, we assess the model using a real-world dataset
180 focused on human motion tracking. The primary experiments are now presented, while additional
181 supporting experiments are given in the appendix.

Table 2: Extrapolation error (expressed as mean±std over the test dataset) evaluated on a test dataset
that possesses the same non-conservative characteristics as the training dataset. All evaluations are
calculated at timestep $k = 35$. The best results are highlighted in **bold** text.

|  | NODE | GLNN | DFLNN (proposed) |
|---|---|---|---|
| Task 1: Damped Double Pendulum | **0.050±0.034** | 0.42±0.19 | 0.051±0.022 |
| Task 2: Dissipative Charged Particle | 0.041±0.015 | $4.0 \cdot 10^7 \pm 1.7 \cdot 10^8$ | **0.0072±0.0029** |
| Task 3: Pixel Pendulum | 0.023±0.025 | 0.022±0.009 | **0.0030 ± 0.0019** |
| Task 4: Human Motion Capture | $2.4 \cdot 10^7 \pm 5.5 \cdot 10^7$ | NaN | **11.75±10.89** |

**Prediction workflow** Unlike training, the prediction workflow involves forward time-stepping. The
183 model is trained by minimizing the ODE residuals of the discrete Lagrange-d'Alembert equations (5)
184 over observed segments to learn $L_\theta$ and $F_\theta$. During the prediction phase, given two initial positions
185 $\{q_0, q_1\}$, the model solves $\mathcal{E}_\Delta(q_0, q_1, \hat{q}_2) = 0$ for $\hat{q}_2$ using the learned $L_\theta$ and $F_\theta$. This process is
186 repeated recursively by feeding predictions as new inputs to generate longer rollouts.

187 To evaluate the models, we compare the *extrapolation error* as the root mean square associated
188 with the $k$-th step for a predicted rollout: Extrapolation Error$_k := \frac{1}{N_\mathcal{T}} \sum_{i=0}^{N_\mathcal{T}} \|q_k^{\{i\}} - \hat{q}_k^{\{i\}}\|_2^2$, where
189 $i$ indexes each trajectory. The true trajectory is the solution at time step $k$ given the same initial
190 condition for $\{q_0, q_1\}$,

191 **Baseline models**   A natural method to compare with is the Generalized Lagrangian Neural Network
192 (GLNN) [3], which models forced Euler-Lagrange dynamics (2) using a finite difference approxima-
193 tion to $\dot{q}$. We also compare with Neural ODE [19], trained directly on position sequences without
194 structural priors. The loss function is equal to the one given in (16). See Appendix C for details.

195 **Learning on learned latent variables from an autoencoder**   We also train a model on high-
196 dimensional, unstructured pixel data $q \in \mathbb{R}^d$ by training an autoencoder to map the latent space
197 $z \in \mathbb{R}^l$ (see Figure 6), where dynamics are modeled. The encoder $\phi_\theta : \mathbb{R}^d \to \mathbb{R}^l$ and decoder
198 $\psi_\theta : \mathbb{R}^l \to \mathbb{R}^d$ are implemented as convolutional networks for image data or feedforward networks
199 otherwise. The autoencoder is trained per timestep separately via the reconstruction loss

$$\mathcal{L}_{\text{AE}}(\phi_\theta, \psi_\theta)(q_n) = \frac{d}{l}\|q_n - \psi_\theta(\phi_\theta(q_n))\|_2^2. \tag{14}$$

200 Allowing the autoencoder to recover meaningful latent coordinates [33], we train the autoencoder
201 simultaneously as the proposed model, weighted by $\omega_{\text{AE}}$, resulting in the complete loss function:

$$
\begin{aligned}
\mathcal{L} = &\frac{\omega_{\text{physics}}}{N_\mathcal{T}(N-1)} \sum_\mathcal{T} \sum_{n=1}^{N-1} \mathcal{L}_{\text{physics}}(L_\theta, F_\theta)\Big(\phi_\theta(q_{n-1}), \phi_\theta(q_n), \phi_\theta(q_{n+1})\Big) \\
&+ \frac{\omega_{\text{reg}}}{R} \sum_{r=1}^{R} \mathcal{L}_{\text{reg}}(L_\theta)\Big(\phi_\theta(q_r), \phi_\theta(q_{r+1})\Big) + \frac{\omega_{\text{AE}}}{N_\mathcal{T}(N+1)} \sum_\mathcal{T} \sum_{n=0}^{N} \mathcal{L}_{\text{AE}}(\phi_\theta, \psi_\theta)(q_n).
\end{aligned}
\tag{15}
$$

202 During prediction, dynamics are computed fully in latent space; the autoencoder is used only to
203 encode initial states and decode the rollout predictions.

204 **Implementation**   The experiments were implemented in Python using the PyTorch framework with
205 the Adam optimizer. See Appendix D for selected hyperparameters and additional computational
206 notes. Code is available at github.

## 4.1   Task 1: Damped Double Pendulum

208 We consider a double pendulum with two identical masses and synthetically generate displacement
209 angle trajectories $\{\theta_1^t, \theta_2^t\}_{t=1}^T$, where $\theta_i^t \in \mathbb{R}$, over $T = 20$ time step and a step size $h = 0.1$. We
210 train on 320 trajectories and evaluate on 10 trajectories. Gaussian noise $\mathcal{N}(0, \sigma^2)$ with variance
211 $\sigma^2 = 10^{-2}h$ is added to each sample to simulate measurement uncertainty.

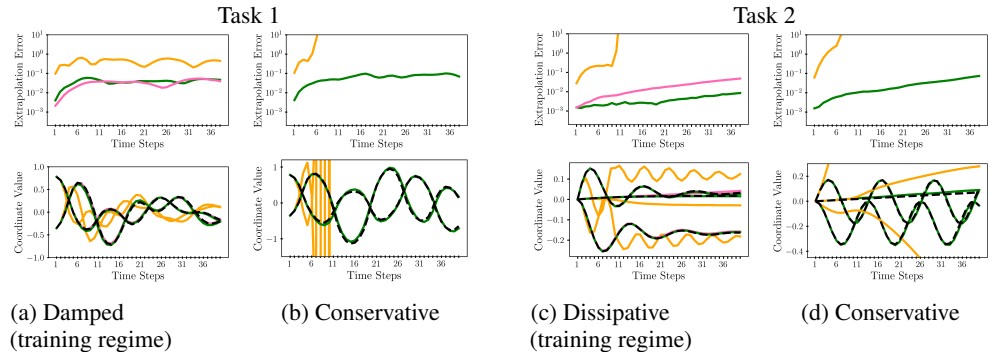

(a) Damped
(training regime)

(b) Conservative

(c) Dissipative
(training regime)

(d) Conservative

Figure 3: Combined results for Task 1 (left) and Task 2 (right). Solid Green lines are DFLNN
(proposed), yellow lines are the GLNN model, and pink lines are a Neural ODE. The ground truth
is indicated with dashed black. (a, c): Rollouts and extrapolation error for the trained model. (b,
d): Turning off the external force component from the learned model to demonstrate the proposed
model's capabilities to distinguish the conserved dynamics (only applicable for DFLNN and GLNN).

212 The performance of the proposed model is shown in Figure 3, with metric evaluations in Table 2.
213 Here, the Lagrangian is assumed to have the form (10) where $U = U(q)$, and the external force
214 is linear (13). The results show that the proposed model achieves comparable performance to the
215 baseline models when applied to a damped system sharing the same characteristics as those present

in the training period. Moreover, when extrapolating onto a conserved regime, the proposed model maintains its accuracy, whereas the baseline models struggle to generalize. Notably, this enables us to make rollouts over a time-span that exceeds the intervals present in the training dataset.

## 4.2  Task 2: Dissipative Charged Particle in a Magnetic Field

Next, we examine the behavior of a charged particle moving through a magnetic field. Our analysis considers a setup characterized by linear dissipation expressed in Cartesian coordinates. The training and test datasets are generated in a similar fashion to that employed in Task 4.1.

For a charged particle traveling in a magnetic field, the potential energy depends on both the position $q$ and the velocity $\dot{q}$. Thus, we assume that the Lagrangian has the form (10) with $U = U(q, \dot{q})$, and the external force is linear (13). The performance of the proposed model is illustrated in Figure 3, with metric evaluations in Table 2. The proposed model shows its superiority over baseline models when rolling out predictions across several time steps, in both non-conservative regimes and when extrapolating to a conservative setting.

## 4.3  Task 3: Pixel data for a Damped Simple Pendulum

To assess the ability of the model to learn on latent embedding from an autoencoder, we use synthetic time series of images simulating a damped simple pendulum. Data are generated using the Gymnasium library [53], a successor to OpenAI's Gym [54], producing $500 \times 500 \times 3$ RGB frames at each timestep. Damping is introduced by modifying the `Pendulum-v1` environment to include a linear dissipative force. All images are cropped around the pendulum, converted to grayscale, and down sampled by a factor of 5. Initial displacements are limited to $\pi/6$ radians. The training dataset comprises 320 trajectories of $T = 100$ time steps each, where $q_t \in \mathbb{R}^{50 \times 30}$. The test dataset contains 10 trajectories. The use of an autoencoder is essential in this setting, reducing the state dimension from $50 \times 30$ to a latent space of dimension ($l = 1$); see Section 4.

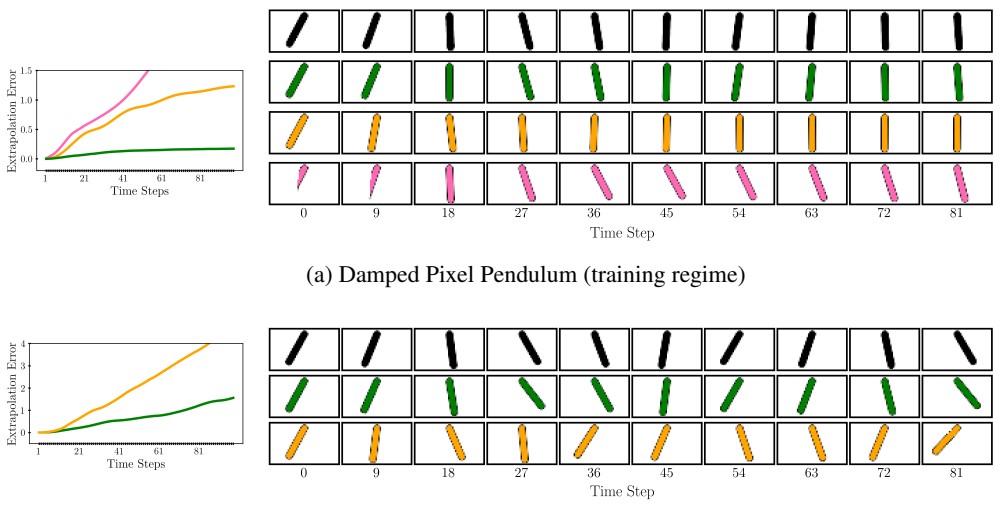

(a) Damped Pixel Pendulum (training regime)

(b) Conservative Pixel Pendulum

Figure 4: Results for a simple pendulum represented through pixel images. Green pendulums are DFLNN (proposed), yellow are the GLNN model, and pink lines are a Neural ODE. The ground truth is indicated with black. (a): Rollouts and extrapolation error for the trained model. (b): Turning off the external force component from the learned model to demonstrate the proposed model's capabilities to distinguish the conserved dynamics (Only applicable for DFLNN and GLNN).

To reduce the impact of autoencoder reconstruction errors during evaluation, we extract structural features using the Harris corner detection method [55]. Detected edges are binarized—assigning 1 to edge pixels and 0 elsewhere—thus focusing the evaluation on shape consistency rather than pixelwise accuracy. Similarity between predicted and ground truth edge maps is quantified using the

243 Dice coefficient [56], which measures overlap between two binary masks. This allows for robust
244 extrapolation error evaluation even when raw pixel reconstruction is imperfect.

245 In Figure 4, we display both the extrapolation error and an example rollout. Metric evaluations are
246 presented in Tabel 2. The findings reveal that the proposed model surpasses the baseline models
247 in predictive rollouts. Additionally, the model exhibits interpretability by extrapolating beyond the
248 training regime to a conserved setting by deactivating the identified external force component.

### 4.4 Task 4: Human Motion Capture

250 We evaluate our method on real-world data of a human swinging from a bar, using motion capture
251 recordings from the CMU Graphics Lab Motion Capture Database [57, 58]. The dataset includes two
252 recordings (subject 43, trials 2 and 3), which we jointly train on. The model is tasked with learning
253 a shared Lagrangian and external force field that captures the dynamics underlying the swinging
254 motion. We model the motion of 10 linked joints on the right side of the body—from the tibia
255 to the radius—including `rtibia`, `rfemur`, `rhipjoint`, `root`, `lowerback`, `upperback`, `thorax`,
256 `rclavicle`, `rhumerus`, and `rradius`. Due to the likely low intrinsic dimensionality of the motion,
257 we apply an autoencoder to reduce the observed dimension $l = 6$.

258 To augment the data, every 10th frame is extracted as a separate trajectory, yielding 10 trajectories
259 per trial— 20 trajectories in total. Joint angles, recorded at 120 Hz relative to a fixed reference, are
260 converted to Cartesian coordinates and smoothed using a Savitzky-Golay filter [59].

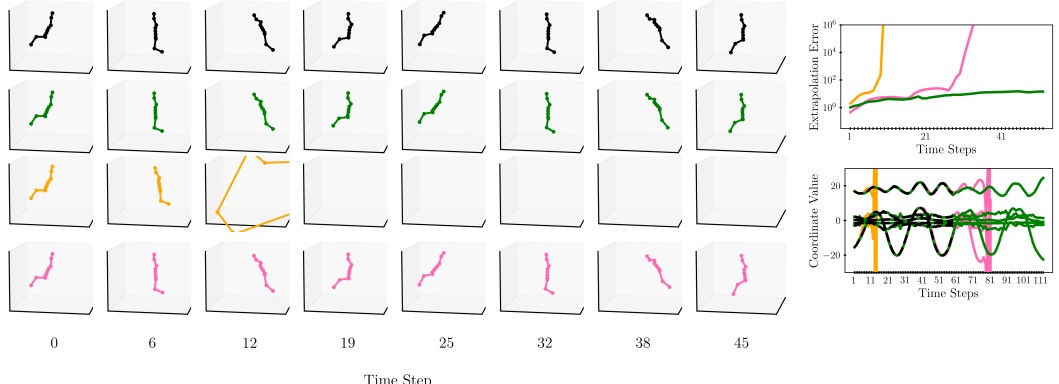

Figure 5: Results and reconstruction (trial 2) of human motion. Green notation is DFLNN (proposed),
yellow is the GLNN model, and pink lines a Neural ODE. The ground truth is indicated in black.
The left panel depicts the full movement represented as a skeletal sketch. The right panel shows the
extrapolation error (top) and the rollout trajectory for the right femur (bottom).

261 We hypothesize that a person swinging from a bar behaves similarly to a multi-pendulum system
262 and aim to identify a Lagrangian with mechanical structure with generalized potential $U(q, \dot{q})$. The
263 external force is assumed to comprise both frictional dissipation and active energy input from the
264 subject. Accordingly, we model it as the sum of a nonlinear dissipative term plus a neural network
265 $F_\theta^{\text{Free}}$ to model the forces exerted by the human motion $F_\theta = -K_\theta(q)\dot{q} + F_\theta^{\text{Free}}$.

266 Figure 5 shows that the model accurately reconstructs the motion, suggesting it has learned general-
267 izable governing equations. When extrapolating beyond the training window, (after $\sim 60$ frames)
268 the proposed model continues to produce plausible trajectories, whereas baseline models diverge
269 significantly earlier.

## 5 Conclusions

271 We have proposed a method for learning mechanical systems from data using the Lagrange-
272 d'Alembert principle. The approach requires the use of position data only, can handle both conserva-
273 tive and dissipative systems, generalizes well when trained and validated on both synthetic as well as
274 real-world data.

## Acknowledgments

The data used in this project was obtained from mocap.cs.cmu.edu. The database was created with funding from NSF EIA-0196217.

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
