# OpenReview forum: "Learning mechanical systems from real-world data using discrete forced Lagrangian dynamics"
_NeurIPS.cc/2025/Conference — Submitted to NeurIPS 2025_

### Official Review · Reviewer_Ynkm · 2025-06-17

**Clarity:** 3
**Significance:** 2
**Originality:** 2
**Rating:** 3
**Confidence:** 4

**Summary:**

The paper proposes a data-driven approach to learn the equation of motions of dynamical systems. The approach addresses the following three challenges:
- Preservation of the variational structure of the dynamics, that is being generated from a principle of least action.
- Trajectory data consisting only of positions, but not velocities or momenta.
- The presence of dissipative forces.
The Authors combine ideas from previous works on Lagrangian Neural Networks, ad well as variational integrators to define a (regularized) loss function to minimize at training time, as well as an autoregressive inference pipelines.

The approach is tested on four datasets of moderate dimension, showing reasonable performance.

**Questions:**

- Can you clarify more concretely what you mean by "structure-preserving" models? I interpreted it as a model that can be linked to a principle of least action, but I don't know if I understood correctly.
- You write: "In practice, velocity and momentum estimates are usually approximated from sequential position measurements using finite differences, which leads to inaccuracies due to noise and truncation errors". Isn't this true for this method as well? At the end of the day, you define $\bar{\dot{q}}\_{n + \frac{1}{2}} = h^{-1}(q\_{n + 1} - q\_{n})$
- In generating the datasets, you wrote: "Training data for these tasks are generated by solving the corresponding analytical ODEs with Gaussian noise applied to each sample." Why the Gaussian noise?

**Ethical Concerns:**

["NO or VERY MINOR ethics concerns only"]

**Final Justification:**

My main concerns are around the originality of the approach, which is a straightforward combination of [a,b]. Nevertheless, I find both the method and its evaluation sound. Furthermore, the Authors did a very good job during the rebuttal phase to address my and other reviewers' comments, clarifying the contributions specific to their work.

Given this year's impossibility to update the manuscript during the rebuttal, however, the Reviewer has no idea how the discussion will be incorporated into a revised version. Many important claims, such as the method not requiring velocity information, were debated at length during the rebuttal, and a fresh perspective on the updated manuscript might be helpful.

[a] Sina Ober-Blöbaum and Christian Offen. Variational learning of euler–lagrange dynamics from data.

[b] J. E. Marsden and M. West Discrete mechanics and variational integrators.

**Limitations:**

Some limitations are discussed in the supplementary material, but they only address the experimental evaluation and not the whole approach.

**Quality:**

3

**Strengths And Weaknesses:**

### Strengths
- The approach is principled and easy to understand. Thanks to automatic differentiation, I guess it is also easy to implement.
- The paper is clear and well-written, with an adequate literature review.
#### Weaknesses
- In my judgement, the paper is a small improvement over the state of the art, essentially packaging ideas from [a, b, c] into a single algorithm. The main contributions, which are stated in Section 1.2, include (a) which is not novel (see Ref [a]), (b) and (c) that are arguably _not_ contributions but features of the model, (d) which is the experimental evaluation and (e) which I think is an original additional experiment but frankly not aligned with the main message of the paper.
- I also have some concerns on the experimental evaluation, and specifically with the lack of the baseline from ref [a]. I actually think that the baselines presented are well chosen:
	- Neural ODE is a general-purpose dynamical model
	- GLNN includes dissipation, but not the variational integrators
	That's why I would have expected the presence of LSI from [a], since in some respects it is complementary to GLNN: includes variational integrators but not dissipation.

Given these comments, I do not recommend the publication at NeurIPS. Despite the paper being well-written and principled, I do not find enough novelty to justify a recommendation of acceptance.

[a] Sina Ober-Blöbaum and Christian Offen. *Variational learning of euler–lagrange dynamics from data.*

[b] Miles Cranmer, Sam Greydanus, Stephan Hoyer, Peter Battaglia, David Spergel, and Shirley Ho. *Lagrangian neural networks.*

[c] J. E. Marsden and M. West *Discrete mechanics and variational integrators.*

---

> ### Author Rebuttal · Authors · 2025-07-30
>
> Dear Reviewer Ynkm ,
>
> We sincerely thank you for your review and constructive feedback. We appreciate your time and the suggestions provided to improve our work.
>
> Below, we address each of your comments and questions:
>
> **Response to W1: The novelty of the paper**
>
> The reviewer is correct that the paper contains ideas related to [a, b, c], and that it is all combined into one algorithm. However, we find it important to emphasize that the suggested model is a significant and practically useful extension, as it can be applied to dissipative systems (unlike [a, b]). Our approach is quite different from LNNs as explained in the paper and in our rebuttal in the answer to reviewer qiZP. We do not learn the Lagrangian to find the Euler-Lagrange equations but to obtain a discrete dynamics which (in absence of external forces) is symplectic. The reference to Marsden and West [c] is of course very relevant. However it has nothing to do with neural networks approximations of Lagrangian and Forces and with finding dynamical systems from data, but rather finding discrete dynamics when Lagrangian and forces are available and given by physical models.
>
> We agree that our approach is similar to that of Ober-Blöbaum and Offen; both our method and theirs are based on discrete variational approaches and use only position data. However, these authors do not consider the discrete Lagrange d'Alembert principle and its discrete counterpart, but only the Hamilton principle. We think that their approach is unsuitable to learn dynamics from real-world data, which is bound to include dissipation effects, e.g., due to friction. Thus, we would argue that the incorporation of external forces, as we are taking into account here, is a highly valuable for real-world applications.
>
> The reviewer commented that (e) is "an original additional experiment but frankly not aligned with the main message of the paper.". We would argue that the experiments are indeed aligned with the main message of the paper, because in this experiment, we are considering dissipative dynamics and demonstrate how the dissipative and non-dissipative parts of the system are being identified by the model. Unlike the two previous experiments, this is all happening in a learned latent space which is included to highlight the reach of the framework. The support of the suggested model to be incorporated with an autoencoder to learn a mapping to a latent space shows a strength of the model, in our opinion. It shows how the measuring technique does not have to be limited to position sensors, but we initiate e.g., video recording. To the best of the authors’ knowledge, learning dissipative dynamics from pixel data in a learned latent space has not been extensively explored in prior work.
>
> [a] Sina Ober-Blöbaum and Christian Offen. Variational learning of euler–lagrange dynamics from data.
>
> [b] Miles Cranmer, Sam Greydanus, Stephan Hoyer, Peter Battaglia, David Spergel, and Shirley Ho. Lagrangian neural networks.
>
> [c] J. E. Marsden and M. West Discrete mechanics and variational integrators.
>
> **Response to W2: Experimental evaluation**
>
> The reviewer highlights the lack of comparison with the method of Ober-Blöbaum and Offen. We are happy to extend the experiments to include a comparison with this method to see how they perform when applied to forced Lagrangian systems in an updated version of the paper. However, we are concerned that this would perhaps be an unfair comparison because their methods were not designed trying to learn also external forces.
>
> What we thought was interesting to investigate instead was learning both external forces and Lagrangian and then reproduce only the conservative dynamics e.g. without forces to see how it would compare to the true conservative discrete dynamics where the forces have been removed. Our experiments show that the Lagrangian is captured correctly. With that said, it would indeed be interesting to see how the structure preserving LSI will compare to the dissipative GLNN (and with our method that meets both structure preserving and dissipative features). We plan to make some experiments and include them in the appendix.
>
> **Response to Q1:**
>
> In this case "structure preserving" means that it satisfies exactly (i.e. to high precision) a discrete Lagrange d'Alembert principle. This is a generalization compared to the discrete Hamilton principle, which does not include external forces. The approach of Ober-Blöbaum and Offen satisfies a discrete Hamilton principle.
>
> **Response to Q2:**
>
> We answered to a similar question to reviewer qiZP (Q1) and to reviewer w3sC (Q1). Please see our answer ro reviewer qiZP, Question 1.
>
> **Response to Q3:**
>
> We have added Gaussian noise to the data to simulate uncertainty and variability to real-world data measurements.
>
> **Response to Limitations:**
>
> We apologize for not fully addressing your concerns about limitations earlier. We will expand the limitations section to also address challenges considering the entire workflow. One of them is the need for a regularization term. This is important to make sure the learned L is regular, and by achieving this, it prevents all of the dynamics from being captured by the force term. This is needed to guide the learned Lagrangian and the learned force to be distinguished from each other. By including such a term, one has to choose the weighting of the different loss terms, which increases the number of hyperparameters. However, this regularisation term is still only guiding the learning, and there is no guarantee that the conservative and non-conservative dynamics to be learned separately. Also, computing such a term is expensive; thus, one has to choose the number of regularisation points carefully to avoid unnecessarily adding computational complexity, which we have already commented on. Moreover, our method is only applicable to problems in which a Lagrangian do exist. This makes the use not possible to apply to all types of system, and requires the user to have some knowledge about the problem at hand.

---

> > ### Comment · Reviewer_qiZP · 2025-08-02
> > **Related Work on Learning Dissipative Dynamics from Pixels in a Learned Latent Space**
> >
> > > To the best of the authors’ knowledge, learning dissipative dynamics from pixel data in a learned latent space has not been extensively explored in prior work.
> >
> > The reviewer (mildly) disagrees with this statement by the authors. This setting has been also explored in prior work, such as in the following references:
> >
> > [Botev et al., 2021] Aleksandar Botev, Andrew Jaegle, Peter Wirnsberger, Daniel Hennes, and Irina Higgins. Which priors mat-
> > ter? benchmarking models for learning latent dynamics. In Thirty-fifth Conference on Neural Information
> > Processing Systems Datasets and Benchmarks Track (Round 1), 2021.
> >
> > [Stölzle et al., 2024] Stölzle, M., & Della Santina, C. (2024). Input-to-state stable coupled oscillator networks for closed-form model-based control in latent space. Advances in Neural Information Processing Systems, 37, 82010-82059.

---

> > ### Comment · Reviewer_Ynkm · 2025-08-04
> >
> > I thank the Authors for the clarifications provided in their response. Let me follow up with a few remarks on my side:
> >
> > About the __novelty and contributions of the paper__, I think we agree that the main _new_ feature of the proposed model is replacing the discrete Hamilton principle used in the work of Ober-Blöbaum and Offen [a] with a discrete Lagrange d'Alembert principle as derived, e.g., in [c]. That being said, and also in light of the interesting discussion between the Authors and Reviewer **qiZP** concerning the approximation of the velocity field, my opinion on the novelty of this submission remains unchanged.
> >
> > About __the experiment on learning dynamics from latents__, I appreciate the explanation from the Authors, and I found their comment about how the "measuring technique does not have to be limited to position sensors" particularly clarifying.
> >
> > About the __experimental evaluation__, my point about comparing to Ober-Blöbaum and Offen was exactly showing the inadequacy of their method when external forces are present. This is not intended to create an unfair comparison to them, but rather to _support_ the relevance of your method. As you have written:
> > > We think that their approach is unsuitable to learn dynamics from real-world data, which is bound to include dissipation effects, e.g., due to friction. Thus, we would argue that the incorporation of external forces, as we are taking into account here, is highly valuable for real-world applications.
> >
> > a baseline as the one I've suggested would justify this claim with factual results. That being said, I am aware of this year's reviewing policies, and the absence of this additional evaluation will not negatively affect my review.
> >
> > About __questions and limitations__, I am satisfied with the Authors' responses.
> >
> > Since the discussion phase is very lively for this paper, I will not update my score just yet. I'll re-evaluate my assessment towards the end of the discussion period, and I will take these couple of remaining days to carefully check the also the other reviewers' rebuttals.

---

> > > ### Author Response · Authors · 2025-08-06
> > >
> > > Dear Reviewer Ynkm,
> > >
> > > We do in fact agree that including a comparison with the LSI method is perhaps exactly what is needed to corroborate our claim that including non-symplectic dynamical terms is useful for more realistic systems. We will therefore update the final version of the paper with the LSI results.
> > >
> > > Furthermore, based on your and qiZP's comments, we believe we should state the novelty of our paper more precisely and why it is interesting for NeurIPS readers. To address this, we will: (1) expand on the related work section to discuss relevant literature on learning dissipative dynamics in latent space such as the articles given by reviewer qiZP; and (2) expand on some future directions that our research enables, including the reseach opportunitunities we have dicussed in the rebuttal to reviwers 9w3C and w3sC.
> > >
> > > (1) Updates to related work section: We believe that a key novelty of our approach that needs to be better emphasised lies in the structured latent representation that allows for modifying the learned latent space dynamics after training. While existing latent-space physics models jointly learn conservative and non-conservative components, to our knowledge, they have not been able to isolate and manipulate these terms independently at inference. In contrast, our method explicitly separates the latent dynamics into an energy-conserving part and forcing term, enabling us to “turn off” the forcing module at inference and thereby simulate undamped system dynamics. Remarkably, we achieve this without ever observing or training on undamped data, demonstrating genuine extrapolation and significantly enhances interpretability. Finally, we validate the practical significance and real-world applicability of this approach through human motion capture data.
> > >
> > > (2) Updates to future direction section: We believe this work could pave the way for future researchers to apply DLNN frameworks to human data in controlled motion problems such as sports biomechanics, physical therapy, as well as robotics, with potential to enhance classification, optimize movement and performance, and plan energy-optimal undamped trajectories from recorded motions to improve efficiency and realism. By combining our approach with high‑dimensional human motion, we are hoping to study the conservative and non-conservative latent representations separately, potentially allowing for more robust and fine-grained motion classification.

---

> ### Author Response · Authors · 2025-08-04
> **Related Work on Learning Dissipative Dynamics from Pixels in a Learned Latent Space**
>
> Thank you for pointing out the related works.
>
> We acknowledge that the cited references also consider dissipative systems. However, our approach differs in a way that we believe provide novel contributions. In particular, our method enforces variational principles directly in latent coordinates, allowing the model to learn a structured, interpretable representation that respects the underlying physics. In doing so, the model explicitly separates the conservative and non-conservative dynamics. Making it possible to remove dissipative dynamics in the learned latent space to reconstruct image sequences governed by the underlying (unobserved) symplectic dynamics.
>
> To the best of our knowledge, this combination of learning in latent space while distinguishing and removing dissipation to recover conservative dynamics has not been extensively explored in prior works.

---

### Official Review · Reviewer_w3sC · 2025-06-24

**Clarity:** 3
**Significance:** 2
**Originality:** 2
**Rating:** 4
**Confidence:** 2

**Summary:**

This paper addresses the problem of learning the dynamics of physical systems directly from position data. It proposes using neural networks to represent the Lagrangian and Lagrangian force functions within the formalism of Lagrangian mechanics. While this is a well-studied problem, previous methods require velocity data, which is often obtained through numerical approximations. Instead, the authors formulate a loss function using a discrete version of the Lagrange–d'Alembert principle. Minimizing this loss generates physically consistent trajectories. The method can also be applied to high-dimensional measurements, such as images, by first mapping them to a latent space using an autoencoder. The authors validate the method on three synthetic data experiments and one real-world data experiment. The experiments suggest that the proposed method outperforms the alternatives considered.

**Questions:**

- The discretized Lagrangian system uses a midpoint approximation to form the generalized velocity coordinates. How is this approximation fundamentally different from using the numerical derivative approximations found in prior work?
- This is a minor point, but the plots, particularly Figure 3, are too small and require significant zooming to be legible.

**Ethical Concerns:**

["NO or VERY MINOR ethics concerns only"]

**Final Justification:**

The authors did address some of my confusions r.e. the velocity approximation, but I still believe the contribution is modest relative to prior work, and the evaluation is somewhat limited. I choose to maintain my weak recommendation that this paper should be accepted.

**Limitations:**

yes

**Quality:**

3

**Strengths And Weaknesses:**

## Strengths

- To the best of my knowledge, the submission is technically sound, and the proposed methodology is reasonable.
- The paper provides supporting evidence for its two main claims from both real-world and synthetic data experiments: that the method is more performant for predicted rollouts, and that the model can separate conservative and dissipative dynamics.
- The paper is clear and generally well-written; the figures effectively communicate the model's structure.
- The mathematical exposition is clear and appropriate for a general audience.
- The proposed method appears to be an advancement on previous work and allows these models to be applied to more general datasets (i.e., from purely positional measurements).

## Weaknesses
- The model is only compared to the GLNN model. It seems that other models could also be suitable for this data, for example, [Dissipative SymODEN](https://arxiv.org/abs/2002.08860) or the Deep Dissipative model proposed in ["Learning Deep Dissipative Dynamics"](https://arxiv.org/abs/2408.11479). Both of these works use similar benchmark problems, so they are comparable to the proposed approach. It is difficult to evaluate the impact of the proposed approach without these comparisons.
- The experiments are somewhat limited and consist mostly of simple synthetic systems.
- The contribution, although significant, appears incremental when compared to existing approaches.

---

> ### Author Rebuttal · Authors · 2025-07-30
>
> Dear Reviewer w3sC ,
>
> We sincerely thank you for your review and constructive feedback. We appreciate your time and the suggestions provided to improve our work.
>
> Below, we address each of your comments and questions:
>
> **Response to W1: Model comparison**
>
> We thank the referee to bring to our attention the two references about SymODEN and "Learning Deep Dissipative Dynamics" and we comment on these methods below.
>
> We agree that the framework of port-Hamiltonian systems used in SymODEN could have similar advantages compared to our method. However, one main difference is the requirement of both position and momentum data for the training (this is a disadvantage similar to the one of requiring velocity data as required in the used baseline GLNN method).
>
> Another observation is that in our motion capturing and pixel example the autoencoder part of the approximation seems to take care of finding the correct coordinates (i.e. both translational and rotational coordinates) which seems to be an advantage similar to what achieved in SymODEN.
>
> As we record snapshots of the data, and the data is discrete in time, we produce approximations that are also discrete and consistent with the data. An important feature of our method is that it is symplectic (when the forces are removed or switched off, i.e. removed after the Lagrangian is learned). This leads to well known advantages when dealing with Hamiltonian systems [a].
>
> We find the second reference "Learning Deep Dissipative Dynamics" indeed very interesting but less similar to our work. The main idea here seems to be the energy preservation/dissipation property (for some Lyapunov function).
>
> For symplectic methods (like ours), backward error analysis guarantees linear error growth and quasi preservation of energy over long times, [a]. While energy dissipation seems to be more relevant for driving a dynamical system to equilibria. The methods of this reference seem to be more useful when learning stable dynamics is crucial. Ours are tested for the moment on problems with relatively small external forces, but we successfully learn both conservative and dissipative forces with our framework. So testing our approach in the context of control systems (e.g. robotics) could be relevant for future work.
>
> We have added references to both approaches in the introductory part of the paper and elsewhere and could add a comment in the conclusion to propose control systems as future work.
>
>
> [a] Hairer, Lubich and Wanner, Geometric Numerical Integration, Springer
>
>
> **Response to W2: Limited experiments**
>
> We partly disagree with this statement. We have used benchmarks that are often used in similar literature (included the two references mentioned by the reviewer). Some more tests are also included in the supplementary material. Test on real-world data and the dissipative pixel pendulum are a strength of our paper and are not usually found in other similar works.
>
> **Response to W2: Limited experiments**
>
> This weakness seems to be related to Question 1 (See answer below).
>
> **Respose to Q1:**
>
> We have answered to a similar question by reviewer 1, qiZP. Please see reviewer qiZP Question 1 for a detailed answer.
>
> **Respose to Q2:**
>
> We will take this comments about  Figure 3 into account when revising the paper.

---

> > ### Comment · Reviewer_w3sC · 2025-08-05
> > **Response to authors**
> >
> > Thanks for the above clarifications, the discussion with reviewer qiZP was enlightening. I have no further questions.

---

### Official Review · Reviewer_9W3U · 2025-07-02

**Clarity:** 2
**Significance:** 2
**Originality:** 3
**Rating:** 4
**Confidence:** 2

**Summary:**

This paper proposes a data-driven framework to learn the equations of motion of mechanical systems based on discrete d’Alembert principle and forced Euler-Lagrange equations. With proper formulation, the proposed system brings the following contributions: (1) it can learn from position data only without the need of velocity or momentum data; (2) it preserves the system structure and is intuitive to incorporate dissipation terms such as external forces.

**Questions:**

I do not have specific technical questions regarding this submission. Given my limited expertise in this area, I will refrain from being overly harsh and currently lean toward acceptance. I look forward to the rebuttal phase to hear perspectives from other reviewers and hope the authors will address the weaknesses and formatting issues I listed.

**Ethical Concerns:**

["NO or VERY MINOR ethics concerns only"]

**Final Justification:**

The rebuttal addressed my concerns. As I'm not an expert in this area, I will keep my original score as borderline accept.

**Limitations:**

yes

**Paper Formatting Concerns:**

- Throughout the manuscript, equations are referenced in an unusual and unclear manner, simply as “(1)” without descriptive labels or names. Please review the LaTeX commands used for equation referencing.
- There are several issues with citations. For instance, Line 93 contains an incorrect citation number ("[421]"), and similarly, Lines 118 and 153 reference "[379]". I recommend carefully proofreading the manuscript to identify and correct these citation issues.
- Table 3 in the supplemental is overfull and exceeds the page margins. Please adjust the formatting to ensure proper layout and readability.

**Quality:**

3

**Strengths And Weaknesses:**

### **Strengths**

I am not an expert in dynamical systems, so I may have missed some mathematical details and did not verify all derivations in detail. Based on my understanding, the main strength of this paper lies in the following:

[A novel theoretical formulation with improved experimental results]
- I feel this work is more like a theoretical paper which presents a theoretically grounded discrete formulation for learning dynamical systems, distinguishing it from many prior methods. While I may not fully understand all aspects of the formulation, the experimental results suggest that the proposed framework is both more stable and more accurate than existing methods, such as Neural ODEs and GLNN.

### **Weaknesses**

[Visualization]
- I’m more familiar with the motion capture stuff. Based on my understanding, the proposed framework can use an autoencoder architecture to learn the underlying dynamic system in the latent space, thereby enabling motion reconstruction and rollout for future motion prediction. To demonstrate this claim, I would say at least a supplemental video should be necessary, including both the ground truth and reconstructed motion sequences and future predictions. The current visualization in Figure 5 is a bit hard to tell the differences, especially between the proposed method (the second row) and Neural ODE (the last row).
- Similarly for the pendulum task, a video demonstration of all the compared methods should be helpful to further show the difference between them.

Also I feel the format of this submission could be improved, which I will list below.

---

> ### Author Rebuttal · Authors · 2025-07-30
>
> Dear Reviewer 9W3U,
>
> We sincerely thank you for your review and constructive feedback. We appreciate your time and the suggestions provided to improve our work.
>
> Below, we address each of your comments and questions:
>
>
> **Response to W1 and W2: Visualization**
>
> We agree with the reviewer that a video would improve the visual comparison across the different approaches, but as far as we understand we are not allowed by NeurIPS policy to use links or pdf files in this rebuttal. Videos will be made available with the code and if possible adding links to the text in the paper.
> In the present version we do include a plot of the errors along with the snapshots of the solution which should give a good comparison of the performance of the various methods.
>
> We do agree with the reviewer when he/she writes " Based on my understanding, the proposed framework can use an autoencoder architecture to learn the underlying dynamic system in the latent space, thereby enabling motion reconstruction and rollout for future motion prediction.  "
> The main contribution of the paper is proposing a method that can learn the dynamics from real-world data and performs well compared to existing methods. Our contribution is not specific to the motion capturing problem. But we believe that testing on real-world data is important, and that learning the forces is essential for good performance on real-world data, simply because these forces are present in the motions we observe in real-world. However in the motion roll out the dissipative forces might be even neglected (at least in some applications). We think our methodology might have good potential and could be explored further in future work in the context of motion capturing and motion manipulation.
>
> (Could add this last observation in the conclusion or future work).
>
> **Response to Paper formatting concerns:**
>
> 1) We will improve this by emphasizing whether we are referring to an equation or figure, etc.: Eq. (1), instead of only (1).
> 2) The number [11, 421] on line 93 refers to the page of the cited reference. This will be clarified [11, p. 421]. Similar for [11, p. 379] on line 118 and 153.
> 3) We will improve the table layout in the updated version.

---

> > ### Comment · Reviewer_9W3U · 2025-08-03
> >
> > Thanks for clarifying! I have no further questions.

---

### Official Review · Reviewer_qiZP · 2025-07-03

**Clarity:** 2
**Significance:** 3
**Originality:** 2
**Rating:** 5
**Confidence:** 3

**Summary:**

This paper proposes a Lagrangian Neural Network (LNN) formulated in discrete time that is also able to learn non-conservative forces, such as dissipation/damping or external forces, from data. The approach is, for example, validated both in state space for learning the motion of a double pendulum, a charged particle in a magnetic field, and human motion capture data, as well as in learned latent space for learning the motion of a pendulum based on image observations.

**Questions:**

- One of my largest doubts about this paper is the difference between the proposed method (that claims that it does not require access to velocity data or finite differences) and vanilla Euler-forward discretizations of existing continuous-time LLNs. Clarifying these aspects would be crucial for my final recommendation on this paper. For example, even though the authors claim that that finite differences for estimating the velocity are not needed, they exactly do that, for example in Equations (7) and (8) via the term $\frac{q_{n+1}-q_n}{h}$. Also, can the authors please prepare a side-by-side comparison of the proposed method with a vanilla Euler-forward discretization of an existing continuous-time LNN (e.g., Lagrangian Neural Networks, or LNNs)? What exactly are the differences during training and rollout?
- Can you please explain in more detail the derivation of Equation (5) and extend the explanation in the manuscript (i.e., show more steps of the derivation)?
- Section 3.3, lines 164-165: Can the method also simultaneously learn external forces and dissipation? If so, how would this be done? If not, why not?
- Section 4.1, line 211: How does the magnitude of the measurement noise compare to the magnitude of the states $q$. Could you show a plot to visualize this?
- My most significant doubt about the results (Section 4.1): It seems very surprising to me that GLNN is so unstable, while all the other methods do not exhibit this issue. Why is this the case? Is this also the case when you remove the measurement noise? Is the GLNN trained without trajectory rollouts? Is the NeuralODE trained with trajectory rollouts? To me, it seems very unintuitive that the GLNN is unstable, while the NeuralODE is stable. Could you please clarify this point further?
- Line 217: Why do the baseline methods struggle to generalize to the conservative regime? Is it because you applied measurement noise to the training data?
- The authors specify that the method was trained in Task 4 on a two trials of human motion capture data. Why did you use not train on more trials, on multiple subjects, or even on the full dataset?

**Ethical Concerns:**

["NO or VERY MINOR ethics concerns only"]

**Final Justification:**

The authors have addressed the reviewer's concerns in the rebuttal, and they have promised several changes, particularly to their claims, for the final version of the paper that should resolve the reviewer's questions and concerns.

Therefore, the reviewer has elevated their score to "5: Accept:", even though the evaluation still seems to some extent limited and there remain concerns that some of the baselines are not evaluated using suitable hyperparameters and/or in suitable settings.

**Limitations:**

The authors claim in the NeurIPS paper checklist that the limitations are discussed in the appendix. However, the appendix is not included in the PDF. Please add the appendix to the PDF. In any case, please also summarize the limitations in the conclusion section of the main text.

**Paper Formatting Concerns:**

Even though the paper is referring to an appendix in the main text, the PDF does not contain any appendix or supplementary material. Please add the appendix to the PDF.

**Quality:**

3

**Strengths And Weaknesses:**

## Strengths

- Reducing the need for velocity data when training LLNs indeed seems like a worthwhile goal, as velocity data is often not available in real-world datasets/systems (without finite differences or integration of IMU data that tends to exhibit drift).
- The method is extensively validated on a variety of tasks, including state space and latent space learning, and both quantitative and qualitative results are strong.

## Weaknesses

- The specific difference of the proposed method (both in terms of rollout/inference, and training loss) with respect to existing continuous-time LLNs trained with Euler-forward discretization is not obvious and should be clarified via a side-by-side comparison.
- Some of the derivation steps are not explained in detail, which makes it hard to follow the derivation of the equations. See the questions section for more details.
- The performance of the baseline methods, specifically the existing LNN/GLNN, seems surprisingly bad and unstable and does not match the performance of existing LNNs in the literature. This should be clarified further. Is it an implementation/hyperparameter issue?

### Detailed Comments on Minor Issues

- Figure 1 could be slightly enlarged to make the text more readable.
- Table 1: Add definitions of the abbreviations of the various methods in the table caption.
- Many of the contributions statements/bulletpoints sound not like actual contributions, but rather like descriptions/features of the method (e.g., b, c). Contributions should claim something novel with respect to the existing literature, not just describe the method.
- For reference [42], the authors should cite the ICLR or IJRR versions of the paper, not the arXiv preprint.
- Table 2: Please report over how many random seeds the statistics (mean and standard deviation) are computed. Also, please directly cite the baseline methods in the table.
- Line 187: The term "Extrapolation error" seems a bit misleading, as it implies generalization to data unseen in the training distribution. Instead, this seems more like a standard "Rollout error"/"Trajectory prediction error".
- Figure 3: Why is panel (a) labeled with "damped" and panel (c) labeled with dissipative? What is the difference between these two terms in this context? Please clarify.
- Lines 266-267: The claim that the method learned "generalizable governing equations" seems questionable as the method was only trained on two trials of motion data of a single subject.

---

> ### Author Rebuttal · Authors · 2025-07-30
>
> Dear Reviewer qiZP ,
>
> We sincerely thank you for your review and constructive feedback. We appreciate your time and the suggestions provided to improve our work.
>
> Below, we address each of your concerns:
>
> **Respone to W1: LNN comparison**
>
> We intend to make a comparison between our approach and LNN and GLNN in the appendix and add comparisons to the Euler-Lagrange equations integrated with forward Euler (LNN+FE). See also our answer to question 1: Q1.
> The major difference between DFLNN and such an approach (LNN/GLNN) is that DFLNN amounts to a symplectic map in the absence of external forces, while this is not the case for LNN+FE. The benefit of this property is that it is well known that symplectic integrators offer superior long-term stability [a].
>
> [a] Hairer, Lubich and Wanner, Geometric Numerical Integration, Springer
>
> **Respone to W2: Derivation**
> Lack of detailed derivation steps.  We will add this to an updated version. See comment below to question 2 (Q2) for the detailed derivations.
>
> **Respone to W3: Performance of baseline models**
>
> To address the reviewer’s comment regarding the performance of LNN/GLNN, we would like to clarify our workflow and implementation of these methods. A key point - which we attempted to highlight in the manuscript and will now emphasize more clearly - is that GLNN is being applied in a context for which it was not originally designed (see Appendix B). The original GLNN paper appears to assume access to velocity measurements, which are not available in our setting. Consequently, our velocity estimates are inherently affected by noise in the position data.
>
> We employed the same training workflow for both our model and the baselines: a two-step prediction followed by backpropagation. Notably, we do not backpropagate through longer rollouts, and this could potentially introduce instability. While this might affect performance, we believe the structure-preserving nature of the proposed DFLNN mitigates such concerns. That said, although we strive for fairness, we cannot rule out the possibility that our implementation might incidentally favor our approach over GLNN. Nonetheless, the training procedure is uniformly applied across all methods and remains relatively naive, constrained by the fact that only position data are available.
>
> **Response to Detailed Comments on Minor Issues:**
> 1,2, 4) This will be adjusted.
> 3) This is a fair point, and we will rephrase the section to separate contributions and features.
> 5) This will be adjusted. We have used 20 different seeds in creating the table.
> 6) We used the term "extrapolation" rather than "prediction" to emphasize that we have only been training on segments of length 3, while the error plots are computed over much longer segments. Also, we used the term to highlight that we are only training on damped datasets, but inference in conservative settings.
> 7) The different use of "damped" and "dissipative" refers to the same mathematical concept. The different use of the termonology was inherent by the natural choice for each physical system. However, using both terms in this setting may be cause confusion, as they essentially describe the same concept. Both will be changed to "dissipative".
>
> **Response to Q1:**
>
> We plan to add a section in the appendix where we compare LNNs and GLNNs to our approach and we list the main differences. If space permits (or in the appendix) we will make also a side-by-side comparison with LNN+FE.
> Here follows a draft comparison of GLNN and DFLNN:
> 1) (G)LNN is approximating the Lagrangian (and Forces) to discover and in turn integrate the (Forced) Euler-Lagrange equations while our method is based on a discrete Lagrange d'Alembert principle (a generalization of the Hamilton variational principle), we do not need to discover the continuous equations and to integrate them numerically, because our method is derived directly from extremizing the discrete variational principle and adding the appropriate discrete forces.
> When the forces are removed our approach is discrete-variational:
> the approach guarantees that when the forces are removed, we are left with a symplectic map (detailed explanations can be found in [a], [b], respecting the qualitative behavior of Hamiltonian systems and guaranteeing long-term stability in Hamiltonian systems (linear error growth [a] and references therein).
> Even though our velocities are approximated by $\frac{q_{n+1}-q_n}{h}$, this is done only to discretize the Lagrange d'Alembert (LA) principle and not to discretize the Forced Euler Lagrange equations. This difference is crucial, because discretizing the Hamilton principle and then extremizing is the key to symplectic integration with variational integrators [b] and it can be seen to correspond to certain, specific symplectic integrators applied to the corresponding Hamiltonian systems under appropriate assumptions (see [a] chapter on Variational Integrators).
> We have not analyzed to which discretization method our method would precisely correspond to when applied to the form of the Euler-Lagrange equations used in LNNs, and under which assumptions the two approaches would become equivalent.
> Generally, a Forward Euler (or any other explicit RK) method applied to integrate the Euler-Lagrange equations is not symplectic.
> 2) GLNN requires the inversion of the Hessian of the Lagrangian wrt $\dot{q}$. This can be an expensive task when this matrix is large and the approach can encounter problems if this Hessian is not invertible or badly conditioned. In fact this could be the reason for the instabilities we see in the experiments for GLNN. See also our answer to Q5 below.
>
> [b] J. E. Marsden and M. West Discrete mechanics and variational integrators.
>
> **Response to Q2:**
>
> The discrete forced Euler-Lagrange equations (eq (5) in our paper) are obtained by taking variations of the action sum (first sum ) in eq (4) and adding the result to the force contributions, second sum in eq (4). Imposing next that the resulting equations vanish for all possible variations leads to eq. (5), [b].
>
> **Response to Q3:**
>
> Yes. This is done in all our experiments and we do it by training two (or more) different networks; one for the Lagrangian and one for the forces.
>
> **Response to Q4:**
>
> Normally in our experiments we learn dynamical systems that are near Hamiltonian systems even if in real-world scenarios there is always some friction and external forces (e.g. the pixel pendulum and the human motion example). This means that we expect the forces to be relatively small compared to the positions in all the experiments.
>
> We will add a plot in the appendix with the forces along with the magnitude of the positions e.g. in the human motion case.
>
> **Response to Q5:**
>
> We understand that we ought to explain more clearly how the experiments are performed.  We plan to add details to section B in the appendix where the features and our use of GLNN is explained. This will be clarified also once our code is made available.
>
> We believe  that the use the inverse-Hessian of L in the numerical integration, without regularization, could lead to instabilities (as we try to invert matrices which are badly conditioned or not invertible). We were not able to find an explanation in the GLNN paper about how the regularization of this Hessian matrix (or of the Lagrangian) is performed.
>
> Also, we found that training the GLNN model presents several challenges. One key issue is the separation between conservative forces, which are expected to be captured by the Lagrangian term, and non-conservative forces, which should be represented by the forcing term. The model architecture does not seem to inherently guarantee that this separation will be learned correctly.
> If this separation is not learned correctly, the learned dynamics will not follow the flow of the (forced) Euler-Lagrange equations. This is not an issue with LNN as it is designed for conservative systems only, unlike GLNN that allows for a dissipative (non-conservtive part). In our investigation of the trained GLNN, we observed that most of the system's dynamics were absorbed into the forcing term, suggesting that the intended separation was not effectively captured. It is possible that a more carefully designed training procedure could improve this behavior, but based on our experiments and evaluation, we find that GLNN requires special treatment to perform as intended. We were not always successful in achieving this separation.
>
> **Response to Q6:**
>
>  Statement line 217: "baseline methods struggle to generalize".
>
> We believe that on shorter time intervals all methods will perform equally well and that most methods are bound to lose performance when the time interval is increased. Our method has a good property of long-term error behavior (and quasi preservation of energy) because it is a symplectic method in absence of external forces and it behaves stably also in presence of dissipative forces and friction. GLNN is not a symplectic method when the forces are switched off.
> However, on very long time intervals also a symplectic integrator might start showing "bad" error growth (e.g. due to propagation of rounding errors).
>
> **Response to Q7:**
>
> As proof of concept, we used this dataset to illustrate the performance of the DFLNN approach compared to other approaches on real-world data, and we did not think it necessary to use large datasets, which would require more extensive experimentation. At this point, we did not have a practical problem in mind, e.g., teaching a robot to move in all sorts of environments and in all sorts of motion, but rather we wanted to show on simple but varied examples how the methodology we propose compares to the baseline methods.
>
> **Response to Paper formatting:**
> Due to the submission format, the appendix is not a part of the main PDF. It is sumbitted in a separate zip file under "Supplementary Material". We apologise for this and hope "Supplementary Material" is accessible for you.

---

> > ### Comment · Reviewer_qiZP · 2025-08-02
> > **Response to Initial Rebuttal**
> >
> > The reviewer thanks the authors for their detailed and constructive response to the initial review. The clarifications provided address most of the concerns raised in the review, and the reviewer appreciates the authors' willingness to improve the paper based on the feedback.
> > While most comments and questions have been addressed satisfactorily, a major issue concerning the claim that no velocity data is needed for the method, as also noted by other reviewers, remains unresolved.
> >
> > > We plan to add a section in the appendix where we compare LNNs and GLNNs to our approach and we list the main differences. If space permits (or in the appendix) we will make also a side-by-side comparison with LNN+FE. Here follows a draft comparison of GLNN and DFLNN:
> > >
> > > 1. (G)LNN is approximating the Lagrangian (and Forces) to discover and in turn integrate the (Forced) Euler-Lagrange equations while our method is based on a discrete Lagrange d'Alembert principle (a generalization of the Hamilton variational principle), we do not need to discover the continuous equations and to integrate them numerically, because our method is derived directly from extremizing the discrete variational principle and adding the appropriate discrete forces. When the forces are removed our approach is discrete-variational: the approach guarantees that when the forces are removed, we are left with a symplectic map (detailed explanations can be found in [a], [b], respecting the qualitative behavior of Hamiltonian systems and guaranteeing long-term stability in Hamiltonian systems (linear error growth [a] and references therein). ...
> >
> > The reviewer appreciates the detailed explanation of the differences between discretizing the continuous Euler-Lagrange equations and the discrete Lagrangian d'Alembert principle. The reviewer thinks that their question/concern is addressed satisfactorily as long as the authors include a similar explanation in the final paper.
> >
> > > Even though our velocities are approximated by $\frac{q_{n+1}-q_n}{h}$, this is done only to discretize the Lagrange d'Alembert (LA) principle and not to discretize the Forced Euler Lagrange equations. This difference is crucial, because discretizing the Hamilton principle and then extremizing is the key to symplectic integration with variational integrators [b] and it can be seen to correspond to certain, specific symplectic integrators applied to the corresponding Hamiltonian systems under appropriate assumptions (see [a] chapter on Variational Integrators)
> >
> > The reviewer is *not* satisfied with this explanation. The authors state frequently both in the manuscript and the rebuttal that the method does *only* require position data, and *not* velocity data. However, as noted also by other reviewers, you implicitly construct this velocity data via finite differences (as many other implementations do as well) and the proposed method might even benefit from having access to less noisy velocity data. The reviewer finds the claim that no access to velocity data is needed and that this is very different from existing methods that use finite differences to estimate the velocity data misleading. The reviewer expects that either the authors provide a more convincing explanation of why the method does not require velocity data, or that they change their claims in the paper of *not* requiring velocity data.
> >
> > > Yes. This is done in all our experiments and we do it by training two (or more) different networks; one for the Lagrangian and one for the forces.
> >
> > I think there is a misunderstanding here. My question was whether the method can learn both external forces and dissipation at the same time, not whether you can train two separate networks for the conservative and external forces. Specifically, it relates to lines 164-165 of the original manuscript, where the authors state _"When prior knowledge suggests a dissipative structure, we instead let Fω be a Rayleigh dissipation
> > function..."_ which implies that the method can either learn external forces or dissipation, but not both at the same time. The reviewer would like to ask the authors to clarify this point in the paper.

---

> > > ### Author Response · Authors · 2025-08-04
> > >
> > > Dear reviewer qiZP,
> > >
> > > We apologize that our initial response did not fully address your concerns, and would like to take this opportunity to clarify further:
> > >
> > > **Comment:**
> > > > Even though our velocities are approximated by $\frac{q_{n+1}-q_{n}}{h}$, this is done only to discretize the Lagrange d'Alembert (LA) principle and not to discretize the Forced Euler Lagrange equations. This difference is crucial, because discretizing the Hamilton principle and then extremizing is the key to symplectic integration with variational integrators [b] and it can be seen to correspond to certain, specific symplectic integrators applied to the corresponding Hamiltonian systems under appropriate assumptions (see [a] chapter on Variational Integrators)
> > >
> > > >> The reviewer is not satisfied with this explanation. The authors state frequently both in the manuscript and the rebuttal that the method does only require position data, and not velocity data. However, as noted also by other reviewers, you implicitly construct this velocity data via finite differences (as many other implementations do as well) and the proposed method might even benefit from having access to less noisy velocity data. The reviewer finds the claim that no access to velocity data is needed and that this is very different from existing methods that use finite differences to estimate the velocity data misleading. The reviewer expects that either the authors provide a more convincing explanation of why the method does not require velocity data, or that they change their claims in the paper of not requiring velocity data.
> > >
> > > We agree that our wording is misrepresenting the key difference between our method and the GLNN. As you correctly point out: the phrasing "requiring velocity" is ambiguous as all methods using finite differences (FD) do not really "require" velocity if the truncation errors due to the FD do not affect the method much. The key difference here lies in the fact that our method naturally incorporates the FD approximation in a way that preserves the variational structure, resulting in a map that is variational/symplectic on $Q \times Q$ as opposed to $TQ$ (for the GLNN). To be explicit: the FD approximation in the GLNN method, takes it from a symplectic integrator on $TQ$ to a non-symplectic integrator on $Q\times Q$.
> > > We will revise the manuscript accordingly. Instead, we will emphasize that the key distinction lies in preserving the variational structure on $Q \times Q$, without introducing extrinsic approximations that break symplecticity, as is the case when FDs are applied post hoc to velocity-dependent formulations. As demonstrated in the numerical experiments, this is advantageous and can otherwise lead to instabilities. We are grateful for the opportunity to clarify this point.
> > >
> > > **Comment:**
> > > > Yes. This is done in all our experiments and we do it by training two (or more) different networks; one for the Lagrangian and one for the forces.
> > >
> > > >> I think there is a misunderstanding here. My question was whether the method can learn both external forces and dissipation at the same time, not whether you can train two separate networks for the conservative and external forces. Specifically, it relates to lines 164-165 of the original manuscript, where the authors state "When prior knowledge suggests a dissipative structure, we instead let $F\omega$ be a Rayleigh dissipation function..." which implies that the method can either learn external forces or dissipation, but not both at the same time. The reviewer would like to ask the authors to clarify this point in the paper.
> > >
> > > In the motion capture experiment, we modelled the total force as the sum of two components: one derived from a Rayleigh dissipation function F_diss and another general state-dependent term F_ext. This setup is motivated by the assumption that the system experiences some form of dissipation, while also allowing the presence of state-dependent external forces. This formulation allows us to learn both dissipation and external forces simultaneously - however, it does not guarantee that they can be perfectly distinguished. For instance, dissipative forces might be partially learned by F_ext. In our experiments this have not been a problem.
> > >
> > > If such separation is important for the task, one could use similar strategy as in [d], and remedied the mixing by penalising the L1 norm of F_ext in the loss function to encourage F_diss to learn the component of the dynamics that can be attributed to a Rayleigh dissipation term. So yes, we can learn both dissipative and external forces simultansously via the two networks F_diss and F_ext, or via just F_ext only.
> > >
> > > [d] Desai, Shaan A., et al. "Port-Hamiltonian neural networks for learning explicit time-dependent dynamical systems."

---

> ### Comment · Reviewer_qiZP · 2025-08-04
>
> Dear authors,
> Thank you for your response. The reviewer is satisfied now with the promised changes by the authors to the manuscript.
>
> > We agree that our wording is misrepresenting the key difference between our method and the GLNN. As you correctly point out: the phrasing "requiring velocity" is ambiguous as all methods using finite differences (FD) do not really "require" velocity if the truncation errors due to the FD do not affect the method much. The key difference here lies in the fact that our method naturally incorporates the FD approximation in a way that preserves the variational structure, resulting in a map that is variational/symplectic on $Q \times Q$ as opposed to $TQ$ (for the GLNN). To be explicit: the FD approximation in the GLNN method, takes it from a symplectic integrator on $TQ$ to a non-symplectic integrator on $Q\times Q$. We will revise the manuscript accordingly. Instead, we will emphasize that the key distinction lies in preserving the variational structure on $Q \times Q$, without introducing extrinsic approximations that break symplecticity, as is the case when FDs are applied post hoc to velocity-dependent formulations. As demonstrated in the numerical experiments, this is advantageous and can otherwise lead to instabilities. We are grateful for the opportunity to clarify this point.
>
> Thank you for the comprehensive explanation. Please make sure to add a similar explanation to the final paper and make sure to **not** claim that your method is (somehow) less reliant on velocities compared to other methods. Indeed, as you explain in your last response, all methods (currently) require access to velocity information one way or another. Either it is available directly, although there are a few sensors that can measure velocity directly in the real world, or it is somehow estimated by finite differences, a Kalman filter, etc. Then, indeed, you can apply different time discretization schemes, and your paper seems to propose an interesting method for preserving symplecticity.
>
> > In the motion capture experiment, we modelled the total force as the sum of two components: one derived from a Rayleigh dissipation function F_diss and another general state-dependent term F_ext. This setup is motivated by the assumption that the system experiences some form of dissipation, while also allowing the presence of state-dependent external forces. This formulation allows us to learn both dissipation and external forces simultaneously - however, it does not guarantee that they can be perfectly distinguished. For instance, dissipative forces might be partially learned by F_ext. In our experiments this have not been a problem.
> >
> > If such separation is important for the task, one could use similar strategy as in [d], and remedied the mixing by penalising the L1 norm of F_ext in the loss function to encourage F_diss to learn the component of the dynamics that can be attributed to a Rayleigh dissipation term. So yes, we can learn both dissipative and external forces simultansously via the two networks F_diss and F_ext, or via just F_ext only.
>
> Ok, that is good to hear! Then, I think lines 164-165 of the original manuscript, where the authors state "When prior knowledge suggests a dissipative structure, we instead let  be a Rayleigh dissipation function..." **should be reworded** as it gives a different impression compared what your method is actually able to do.

---

> > ### Author Response · Authors · 2025-08-04
> >
> > Dear reviewer qiZP,
> >
> > Thank you for your constructive review and helpful comments. We appreciate the time and effort you put into evaluating our work. We will incorporate all the suggested changes into the final version of the paper. We believe these changes will significantly improve the clarity and accuracy of the paper, and we thank you again for your valuable feedback.

---

### Note · Authors · 2025-08-13

Dear reviewers and AC,

We thank the reviewers for the time and discussions devoted to our work. We believe that the scientific concerns have been addressed and the updates resulting from this discussion will significantly enhance the clarity, rigour, and impact of our work. We will now use this opportunity to summarise these concerns and how they are addressed in the final version.

**Velocity data claim.** Several reviewers noted our misleading claim about using finite differences (FD) in our approach compared to the continuous time approaches like LNN/GLNN. We will clarify that although all methods require FDs, our method naturally incorporates the FD approximations in a way that *preserves the variational structure*, resulting in a map that is variational/symplectic *in discrete time*. Unlike the continuous time approaches, where FD approximations break this structure. A new section will explain this important point, based on the numerous productive discussions (in particular with qiZP).

**Novelty framing.** We will state the novelty of our paper more precisely by: (1) expanding on the related work section to discuss relevant literature on learning dissipative dynamics in latent space such as the articles given by qiZP as well as improve the clarity of the comparisons between other methods (see qiZP’s Q1 response); and (2) expand on interesting future directions that our research enables in interdisciplinary fields, (see responses to 9w3C, w3sC and latest response to Ynkm).

**Baseline performance.** We will include additional details in the appendix to explain the GLNN implementation and discuss how mistreating FD errors can lead to instabilities. Further clarification will be provided upon the release of our code. See also the rebuttal to reviewer qiZP Q5.

**New baseline comparisons.** As suggested by qiZP and Ynkm, we will include side-by-side comparisons to LNN+FE and LSI to further support our claim that incorporating forcing is beneficial for realistic systems.

**Learning dissipative and external forces.** We will clarify the model’s ability to learn dissipation and forces simultaneously. Additionally, we will visualise the forces in the human motion experiment (see Q3 and Q4 of qiZP).

**Visualization.** We will add videos to our GitHub’s README.

**Minor issues** regarding clarity, formatting, and layout will be addressed. We will also add the suggested references.

---

### Decision · Program_Chairs · 2025-09-17

**Decision:**

Reject

**Comment:**

The authors propose a method to learn the dynamics of a system using only positional measurements, while allowing for the presence of dissipative forces. The core idea leverages a discrete version of the Lagrange–d’Alembert principle, uses neural networks to represent the Lagrangian and Lagrangian force functions, while minimizing the proposed loss ensures that the generated trajectories are physically consistent. The efficiency of the approach is demonstrated on several synthetic and real-world tasks.

The paper received mixed scores when considering the confidence, though reviewers generally acknowledged that it provides a meaningful contribution. In particular:
- The idea of learning dynamics using only position data is appealing, though it is considered somewhat incremental relative to prior work.
- The paper is generally well-written and accessible, although some reviewers requested additional clarifications and explanations.
- The technical background appears sound, the methodology reasonable, and the empirical validation reasonable. However, additional empirical and analytical comparisons would strengthen the paper.

Some concerns raised during the review were addressed in the rebuttal, with the most important regarding the depth of the empirical validation, the potential novelty, and consequently the overall significance of the paper. The reviewers also raised concerns about how the clarifications and additional results promised in the rebuttal could be effectively integrated into the manuscript, and whether such integration would necessitate another round of review.

Despite its merits, the significance of the approach ultimately depends on the empirical results, and it also appears that many updates are required, as confirmed by the authors, suggesting that a fresh round of review may be necessary to reassess the paper. I therefore recommend rejection and encourage the authors to consider the feedback and improve the manuscript accordingly.